# Parameters Identification of High Temperature Damage Model of X12 Alloy Steel for Ultra-Supercritical Rotor Using Inverse Optimization Technique

**DOI:** 10.3390/ma14030695

**Published:** 2021-02-02

**Authors:** Xuewen Chen, Kexue Du, Yuqing Du, Tingting Lian, Jiqi Liu, Rongren Bai, Zhipeng Li, Yisi Yang, Dongwon Jung

**Affiliations:** 1School of Materials Science and Engineering, Henan University of Science and Technology, 263 Kaiyuan Avenue, Luoyang 471023, China; 18437955329@163.com (K.D.); DuyqStephanie@163.com (Y.D.); lian_tingting@126.com (T.L.); 18848970317@163.com (J.L.); bairongren317@163.com (R.B.); mrlzp597@163.com (Z.L.); 15838510562@163.com (Y.Y.); 2Faculty of Mechanical, Jeju National University, 102 Jejudaehak-ro, Jeju-si 63243, Korea; jdwcheju@cheju.ac.kr

**Keywords:** X12 alloy steel, fracture strain, high temperature damage model, parameters identification, stress triaxiality

## Abstract

X12 alloy steel is a new generation material for manufacturing ultra-supercritical generator rotors. Cracks will appear on the forgings during the forging process and the rotors will be scrapped in serious cases. To optimize the forging process of the rotor and avoid the occurrence of crack defects in the hot forming process, based on Oyane damage model, a high temperature damage model of X12 alloy steel was proposed by introducing the influences of temperature and strain rate on the damage evolution. A reverse analysis method was proposed to determine the critical damage value of Oyane damage model by comparing experimental and simulated fracture displacement in the tensile tests. Then, the critical damage value was determined as a function of temperature and strain rate. The high temperature damage model was combined to the commercial finite element software FORGE^®^ to simulate the high temperature tensile test. The accuracy of the damage model was verified by comparing the difference of the fracture displacement between simulated and experimental samples. Additionally, as stress triaxiality is a significant factor influencing the damage behavior of ductile materials, the effects of temperature and strain rate on the stress triaxiality of X12 alloy steel was analyzed by simulating the high temperature tensile process, and the damage mechanism of X12 alloy steel under high stress triaxiality was analyzed by SEM (Scanning Electron Microscope).

## 1. Introduction

With the improvement of power generation efficiency, the working temperature and pressure of steam turbines and unit capacity have also been continuously improved. Thermal power units have also a development from the initial critical, subcritical (steam pressure 16 MPa, steam temperature 538 °C), supercritical (steam pressure 24 MPa, steam temperature 566 °C) to ultra-supercritical (steam pressure 26 MPa, steam temperature 600 °C), the power supply efficiency has increased from 39% to 45% [1]. Under the long-term action of this high temperature and high pressure working environment, the rotors will creep and have a reduction in service life. In order to improve the service life of the rotor, the rotors are required to have an excellent performance in high temperature. X12(X12CrMoWVNbN10-1-1) alloy steel is used in ultra-supercritical generating sets because of its comprehensive mechanical properties, such as good strength, toughness, and corrosion resistance at high temperature conditions [2,3]. It usually takes hundreds or even tens of thousands of hours for rotors to undergo creep deformation in the service process while the deformation time is short in the forging process [4]. Temperature and strain rate are important factors affecting the high-temperature rheological properties of the material during the forging process [5]. During the process of hot forging, the cracks on the surface or internal part will be caused by its unreasonable design. When it is used in practice, the rotor would be scrapped, resulting in serious economic loss. Therefore, in order to effectively prevent the forging crack defects of ultra-supercritical high and medium pressure rotor and improve the quality of final forgings, it is of great significance to study the crack formation mechanism of X12 alloy steel under hot forging process and establish a damage model considering temperature and strain rate [6].

In recent decades, many scholars have proposed corresponding damage models by studying the mechanism of crack formation and the damage behavior of materials. Freudenthal [7] thought that when the strain energy of per unit volume reached the critical value, macrocracks would occur in the material and the Freudenthal damage model was firstly proposed from the energy point of view. Cockcroft and Latham damage model was put forward by Cockcroft and Latham [8] and it was considered that the fracture of the material was mainly caused by principal tensile stress. Based on the Cockcroft and Latham damage model, Brozzo [9] proposed the Brozzo damage model by considering the influences of hydrostatic pressure on damage. Oh and Kobayashi [10] proposed the Normalized Cockcroft and Latham (NCL) damage model based on the cavity growth theory and considered the ratio of the maximum principal stress to the equivalent stress. Rice and Tracey [11] studied the ductile damage process with isolated spherical defects under three-dimensional stress and described the mechanical behavior and geometric characteristics of the fracture in the process of material damage, then the Rice and Tracey damage model was proposed. Oyane [12] believed that the voids were caused by large deformation inclusions or the second phase particles, the Oyane damage model considering stress triaxiality was proposed. Gurson [13] derived a damage model describing the nucleation, growth, and aggregation of voids by taking the void volume fraction *f* as a damage variable. Tvergaard and Needleman [14] proposed a classical GTN damage model by considering the influences between adjacent cavities based on the Gurson model. Lemaitre [15] proposed the Lemaitre damage model on the basis of the continuous damage theory. Bouchard [16] modified the Lemaitre damage model by considering the void closure effect, making it suitable for high plastic materials. Bonora [17] proposed the Bonora damage model based on the continuous damage mechanics framework and the damage model proposed by Lemaitre.

Although many damage models have been proposed and used widely, their limitations lie in that the damage evolution of materials during high temperature deformation cannot be predicted directly by the cold forming damage model. In the process of hot forging, temperature, and strain rate are two important factors affecting the internal damage evolution of materials. During the process of hot forging, the flow stress and tensile strength of the material will decrease with the temperature increasing. With the increase of temperature, the grain growth will be promoted, and the thermal activation energy will increase. In the process of material deformation, the hardening of the material will intensify as the temperature rises. Dislocation slip, phase transition, and dynamic recrystallization/recovery are all affected by deformation temperature and strain rate. To predict the damage behavior of materials during high temperature deformation, many scholars have done researches. The initial attempt to extend the damage model of cold deformation to hot deformation was the classic Johnson Cook model, it considered temperature and strain rate [18]. Khan and Liu [19] proposed a damage model by considering the influences of temperature and strain rate, and it can predict the damage behavior of isotropic and anisotropic materials at high temperature. By modifying the Oyane-sato criterion, Novella [20] established a model to predict the damage of AA6062-T6 aluminum alloy during cross wedge rolling at high temperature. On the basis of the NCL damage model, Chen et al. [21] introduced the effects of temperature and strain rate on damage evolution and established a high temperature damage model of 45Cr4NiMoV steel.

An accurate identification of damage model parameters is the basis of accurately predicting cracks formation in material forming process by finite element simulation. By studying the change rule of the parameters related to the damage state, the critical damage values of material can be indirectly determined. Commonly used methods are elastic modulus method, density method, void volume method and so on. The limitations of these methods are that numerous repeated tests are needed to reduce errors, and different methods are selected for different material models. In addition, with the temperature increasing, the plasticity of the material increases, making it difficult to identify some parameters related to damage model accurately (such as elastic modulus, void ratio, etc.). Parameter reverse methods are based on some physical quantities which are easy to measure, such as the fracture displacement and the displacement load curve of samples. By selecting an appropriate optimization algorithm, the parameters are adjusted continuously until the optimization target meets the accuracy requirements. Tang B T [22] determined the Lemaitre damage model parameters of 22MnB5 steel during the hot stamping process by comparing the experimental and simulated displacement load curves. Wang [23] determined the four material parameters of the GTN model by using the response surface method and the least square method and studied the damage and cracking behavior of high-strength steel BR1500HS at 20–800 °C. M. Abbasi [24] determined the GTN model parameters of IF steel by combining the reverse method and response surface method. By comparing the forming limit diagram obtained by finite element simulation with by experiment, M. Abbasi verified the accuracy of the obtained GTN model parameters.

Based on the above analysis, the influences of temperature and strain rate on the damage behavior of materials have been considered, a high temperature damage model of X12 alloy steel was proposed on the basis of the Oyane damage model. The tensile tests of X12 alloy steel samples at 950–1150 °C and strain rates of 0.1–5 s^−1^ were executed by Gleeble-1500D thermo-mechanical simulator. Combining optimization algorithm, finite element method and physical experiments, a reverse analysis method was proposed to determine the parameters of X12 alloy steel high temperature damage model. By comparing the fracture displacement of the specimen obtained from simulation of high temperature tensile test with the experimental fracture displacement, the veracity of the proposed damage model is verified. Stress triaxiality (the ratio of the hydrostatic pressure to the equivalent stress) is a significant factor influencing the damage behavior of ductile materials [25]. When the material is in a state of tensile stress (the stress triaxiality is greater than 0), it will promote the generation and propagation of cracks; When the material is under the state of compressive stress (the stress triaxiality is less than 0), the generation and propagation of cracks will be restrained and the cavity will be closed, thereby improving the bearing capacity of the material. In order to reveal the effects of temperature and strain rate on the change of stress triaxiality in the tensile process. In this paper, the variations of stress triaxiality under different deformation conditions were researched and the damage mechanism of X12 alloy steel at elevated temperature was analyzed.

## 2. Materials and Methods

X12 alloy steel was used in the experiment, and its chemical composition has been shown in Table 1. The tensile test at elevated temperature was executed on the Gleeble-1500D thermal-mechanical simulator, the testing temperatures are 950 °C, 1000 °C, 1050 °C, 1100 °C, and 1150 °C; the strain rates are 0.1 s^−1^, 0.5 s^−1^, 1 s^−1^, and 5 s^−1^. To obtain the temperature distribution along the length of the sample, two thermocouples were added at 7.5 mm and 17.5 mm from the center of it. The shape of the sample and the position of the thermocouples have been shown in Figure 1. The experiment conditions have been shown in Figure 2. The sample was heated to the specified temperature at 5 °C/s, then it was executed after socking for 3 min. After the sample fractured, the sample was water cooled, then SEM was used to scan the fracture surface of the sample.

## 3. Analysis of Experiment Results and Constitutive Model of X12 Alloy Steel

### 3.1. Analysis of Experiment Results

Figure 3 is the temperature distribution under the deformation condition of 1000 °C. It can be seen that only the temperature at the center of the sample reaches the preset temperature, the temperature far away from the center drops approximately along the quadratic function. The results show that the temperature distribution along the length of the sample is not uniform. The true stress–strain curves of X12 alloy steel at different temperatures and strain rates have been shown in Figure 4. It can be seen that there are three deformation zones in the tensile process of X12 alloy steel: elastic deformation zone, plastic deformation zone, and necking zone. The stress–strain relationship does not satisfy the uniaxial stress state when the necking occurred; the stress–strain relationship between elastic deformation zone and plastic deformation zone still satisfies the uniaxial stress state. However, the elastic deformation zone of X12 alloy steel is very small at high temperature compared with plastic deformation, it is difficult to observe the elastic deformation zone on the curve. Taking 1150 °C, 1 s^−1^ as an example, elastic deformation only accounts for 0.429% of the entire deformation process. In the process of tensile deformation of X12 alloy steel, the strain rates are also different in the three stages. Taking 1150 °C, 1 s^−1^ as an example, the strain rate of X12 alloy steel in the elastic deformation stage is 0.459 s^−1^ and in the plastic deformation stage is 0.922 s^−1^; the strain rate of X12 steel in the necking stage can be approximated by the average method ($\dot{\varepsilon }=\Delta \varepsilon /\Delta t$) as 12.890 s^−1^. It can be seen that the strain rate only in the plastic deformation process is close to the preset strain rate in the entire tensile deformation process. As the proportion of the elastic deformation stage in the entire deformation process is very small, the stress and strain data of the necking stage do not meet the uniaxial stress state, which makes them unusable; therefore, the strain rate in the plastic deformation stage can be basically used as the strain rate in the elastic and plastic deformation stage. Although the strain rate in the plastic stage is different from the preset one and their differences are very small. Taking 1150 °C, 1 s^−1^ as an example, the difference between them is only 0.078 s^−1^. It obvious that the plasticity of X12 alloy steel at 1050–1150 °C is better than that at 950–1000 °C. Under the condition of a constant strain rate, the flow stress of X12 alloy steel decreases with the increase of temperature. The flow stress of X12 alloy steel increases with the increase of strain rate under the condition of a constant temperature. When the temperature is lower than 1000 °C, the flow stress of X12 alloy steel increases rapidly, and the peak strain corresponding to the peak stress is smaller. As the temperature increases, the peak flow stress of X12 alloy steel also decreases. When the temperature exceeds 1000 °C, its flow stress gradually increases. The flow stress of X12 alloy steel decreases when the material is necked. The reason is that V, Nb, and other alloying elements in X12 alloy steel form compounds with C and N. These compounds are dispersed on the austenite grain boundaries in the form of second phase particles, which hinder the movement of austenite grain boundaries and dislocations, resulting in the increase of strength and the decrease of plasticity of X12 alloy steel. With the temperature increasing, the second phase particles gradually dissolve and the “pinning” effect on the austenite grain boundary weakens. It reduces the strength of the X12 alloy steel and increases its plasticity, which is beneficial to the movement of dislocations [26].

The fracture strain $\varepsilon_{f}$ of X12 alloy steel samples can be obtained by Equation (1) [27]:(1)$$\varepsilon_{f}=2\ln\left(\frac{a_{0}}{a_{f}}\right)$$
where $a_{0}$ and $a_{f}$ are the original diameter of the tensile specimens and the diameter of the specimens after fracture respectively.

Figure 5 is a graph showing the relationship between fracture strain of X12 alloy steel and temperature and strain rate. As we can see, with the increase strain rate, the fracture strain of X12 alloy steel increases at first and then decreases at an invariant temperature. When the strain rate is equal to 1 s^−1^, the fracture strain reaches the maximum. Taking 1000 °C as an example, when the strain rate increases from 0.1 s^−1^ to 1 s^−1^, the fracture strain of X12 alloy steel increases from 1.9959 to 2.6899; as the strain rate increases, the fracture strain of X12 alloy steel decreases from 2.6899 to 2.3266. Overall, The fracture strain of X12 alloy steel increases with the increase of temperature gradually when the strain rate is a constant. Taking 1 s^−1^ as an example, when the temperature increases from 950 °C to 1150 °C, the fracture strain of X12 alloy steel increases from 1.8861 to 3.4364. However, X12 alloy steel has the maximum fracture strain when the temperature is 1050 °C and the strain rate is 0.1 s^−1^. When the strain rate is 0.1 s^−1^, the temperature increases from 950 °C to 1050 °C, the fracture strain of X12 alloy steel increases from 1.5702 to 3.4667; the temperature increases from 1050 °C to 1100 °C, the fracture strain of X12 alloy steel decreases from 3.4667 to 2.9808; the temperature increases from 1100 °C to 1150 °C, the fracture strain of X12 alloy steel increases from 2.9808 to 3.1157. When the temperature is 1050 °C, the fracture strain of X12 alloy steel decreases from 3.4667 to 2.7045 with the increase of strain rate from 0.1 s^−1^ to 0.5 s^−1^; when the strain rate increases from 0.5 s^−1^ to 1 s^−1^, the fracture strain of X12 alloy steel increases from 2.7045 to 2.9221; when the strain rate increases from 1 s^−1^ to 5 s^−1^, the fracture strain of X12 alloy steel decreases from 2.9921 to 2.4166.

### 3.2. X12 Alloy Steel Constitutive Model

The constitutive model of material is the basic material parameter for finite element simulation analysis. In this paper, the Hansel–Spittel constitutive model was used to describe the tensile rheological behavior of X12 alloy steel [28]:(2)$$\sigma =Ae^{m_{1}T}\varepsilon^{m_{2}}{\dot{\varepsilon }}^{m_{3}}e^{m_{4}/\varepsilon }$$
where ε is the equivalent strain, $\dot{\varepsilon }$ is the equivalent strain rate, *T* is the deformation temperature, and *A* and m_1_~m_4_ are the material parameters. The material parameters in the Hansel–Spittel constitutive model can be obtained by nonlinear fitting to the stress–strain data in Figure 4. The results are shown in Table 2.

## 4. High Temperature Damage Model and Parameters Identification of X12 Alloy Steel

### 4.1. High Temperature Damage Model of X12 Alloy Steel

The physical observation and micromechanical analysis of ductile fracture have promoted the development of ductile damage models. On the basis of the principle of continuous pores in materials, Oyane established an Oyane damage model suitable for porous materials. The effect of stress triaxiality on damage is considered in this model.
(3)$$D=\int_{0}^{{\bar{\varepsilon }}_{f}}\left(1+B\frac{\sigma_{H}}{\sigma_{eq}}\right)d\varepsilon$$
where *B* is the material damage parameters, *B* = 3 [29]; $\sigma_{H}$ is hydrostatic pressure; $\sigma_{eq}$ is equivalent stress. When the damage value D exceeds the critical damage value $D_{c}$, the material fracture.

By analyzing the relationship between fracture strain and temperature and strain rate in Figure 5, the conclusion reached is that temperature and strain rate affect the fracture strain of X12 alloy steel in hot forging process. However, the Oyane damage model cannot predict the occurrence of cracks accurately when applied to the hot forging process because the influences of temperature and strain rate on damage are not considered. To overcome this limitation, the Oyane damage model was modified by considering the influences of temperature and strain rate, and a more accurate damage model of X12 alloy steel under high temperature conditions was established. This enables the modified Oyane damage model to describe the damage behavior of materials under the hot forging process.
(4)$$D_{\mathrm{new}}=\int_{0}^{{\bar{\varepsilon }}_{f}}f\left(T,\dot{\varepsilon }\right)\left(1+B\frac{\sigma_{H}}{\sigma_{eq}}\right)d\varepsilon$$
where $f\left(T,\dot{\varepsilon }\right)=\frac{1}{D_{c}\left(T,\dot{\varepsilon }\right)}$, $D_{c}\left(T,\dot{\varepsilon }\right)$ is a function of the critical value $D_{c}$ of the Oyane damage model with the temperature and strain rate. Temperature and strain rate were considered in Z ($Z=\dot{\varepsilon }\exp(Q/RT)$) parameter, the influences of temperature and strain rate on damage was introduced through Z parameter.

### 4.2. Identification of Parameters of X12 Alloy Steel High Temperature Damage Model

Through the combination of experiment and finite element method, the Oyane critical damage value $D_{c}$ of X12 alloy steel at various temperatures and strain rates can be obtained. In this paper, the idea of reverse analysis method is to determine the damage parameter $D_{c}$ as the optimization parameter, the optimization algorithm is chosen as the identification method to minimize the optimization objectives. At the same time, the tensile tests at elevated temperature were simulated by the FORGE^®^ (Paris, France). The fracture displacement, which was obtained from the simulation and the experiment respectively, was compared to evaluate the error, the damage parameters corresponding to the minimum error were the optimal solution. Genetic algorithm (GA) is a kind of random search method derived from the evolution law of organisms. It has high robustness and good convergence. The complex conundrums which are too tough to traditional optimization algorithms can be figured out by GA with effect. Therefore, GA was selected as the optimization algorithm to determine the parameters of damage model in this paper. The objective function is given by Equation (5), the reverse analysis process is shown in Figure 6. To improve the accuracy of calculation and reduce the calculation time, the local mesh refinement technology was adopted. A smaller mesh with a size of 0.1 mm was used in the deformation zone; the size of the transition region is 0.2 mm, located between the refined and non-refined regions, and the other mesh size is 1 mm. Figure 7 shows the finite element model mesh and the temperature distribution along the length of the sample at 950 °C. The parameters of the Hansel–Spittel constitutive model used in the simulation are shown in Table 2.
(5)$$Q={\left(y^{\mathrm{simu}}-y^{\exp}\right)}^{2}$$
where $y^{\mathrm{simu}}$, $y^{\exp}$ are the fracture displacements of X12 alloy steel samples obtained by simulation and by experiment, respectively. Table 3 shows the optimized critical damage values $D_{c}$ of the Oyane damage model at various temperatures and strain rates.

Through regression analysis of the data in Table 3, the evolution law of critical damage value at high temperature was obtained. The damage model of X12 alloy steel at high temperature was established.
(6)$$\left\{\begin{array}{l} D_{\mathrm{new}}=\int_{0}^{{\bar{\varepsilon }}_{f}}f\left(T,\dot{\varepsilon }\right)\left(1+B\frac{\sigma_{H}}{\sigma_{eq}}\right)d\varepsilon \\ f\left(T,\dot{\varepsilon }\right)=1/\left(8.003\sin\left(0.0817\ln Z+11.02\right)+0.2246\sin\left(0.9267\ln Z-21.0\right)\right) \end{array}\right.$$

When $D_{\mathrm{new}}$ is greater than 1, the fracture occurs. Figure 8 shows the curve of the relationship between lnZ and the critical damage value $D_{c}$. It can be seen that with the increase of the Z parameter, the critical damage value $D_{c}$ gradually decreases, indicating that the damage evolution at high temperature is affected by temperature and strain rate.

### 4.3. Validation of X12 Alloy Steel High Temperature Damage Model

To verify the established damage model of X12 alloy steel at elevated temperature, it was combined to finite element software FORGE^®^. According to the elevated temperature test results simulated, the veracity of the damage model at elevated temperature was verified by contrasting the fracture displacement between simulated and experimental ones.

The sample after fracture is shown in Figure 9. L is the original length of the sample, $L_{e}$ is the fracture length of the specimen obtained from the experiment, and $L_{s}$ is the length of the specimen obtained by simulation. The fracture displacement of sample ($\Delta L_{s}$) in the simulation and the fracture displacement of sample ($\Delta L_{e}$) in the experiment can be obtained by the following formulas:(7)$$\Delta L_{e}=L_{e}-L$$
(8)$$\Delta L_{s}=L_{s}-L$$

Figure 10 is the graph of the relative error of the fracture displacement. It can be seen that the relative error under each experimental condition is very small. The minimum error is 0.058%, and the maximum error is 12.925% which occurs at 1050 °C and strain rate of 0.1 s^−1^. The remaining errors are within 7%, the average error is only 3.312%. Therefore, the numerical simulation results have a good fit with the experimental results. Figure 11 shows the correlation diagram of fracture displacement at different temperatures and strain rates. The closer the correlation coefficient R is to 1, the smaller the root mean square error (RMSE) and residual sum of squares (RSS) are, the more accurate the prediction results of the model are. According to Equation (9), R is 0.992, RMSE is 0.607, and RSS is 7.371, which mean that the model has a high prediction accuracy. Fracture displacement error analysis and correlation both analysis show that the high temperature damage model of X12 alloy steel has a high prediction accuracy and credibility, it can be used to predict its fracture behavior of during high temperature deformation.


(9)$$\begin{array}{l} R=\frac{\sum_{i=1}^{N}\left(\Delta L_{e}^{i}-{\bar{\Delta L}}_{e}\right)\left(\Delta L_{s}^{i}-{\bar{\Delta L}}_{s}\right)}{\sqrt{\sum_{i=1}^{N}{\left(\Delta L_{e}^{i}-{\bar{\Delta L}}_{e}\right)}^{2}}\sqrt{{\sum_{i=1}^{N}\left(\Delta L_{s}^{i}-{\bar{\Delta L}}_{s}\right)}^{2}}}\\ RSS=\sum_{i=1}^{N}{\left(\Delta L_{s}^{i}-\Delta L_{e}^{i}\right)}^{2}\\ RMSE=\sqrt{\frac{1}{N}\sum_{i=1}^{N}{\left(\Delta L_{s}^{i}-\Delta L_{e}^{i}\right)}^{2}} \end{array}$$


### 4.4. Effect of Stress Triaxiality on Material Mamage and High Temperature Damage Mechanism of X12 Alloy Steel

To study the change of stress triaxiality during deformation, the point tracking technology provided by FORGE^®^ was used to track the change of stress triaxiality at the center point of the specimen. The setting of the tracking point is shown in Figure 7. It can be seen that the stress triaxiality is always greater than 0 during the tensile process. Figure 12a shows the variation of equivalent strain and stress triaxiality at different temperatures at strain rate of 0.1 s^−1^. It can be seen that when the strain rate is unchanged, the stress triaxiality increases with the increase of equivalent strain. When the temperature is lower than 1000 °C, the change rate of stress triaxiality increases rapidly. When the temperature exceeds 1000 °C, the change rate of stress triaxiality gradually slows down with the increase of temperature, a plateau appears, indicating that the plasticity of X12 alloy steel increases with the temperature increasing. Figure 12b shows the triaxiality-damage diagram at different temperatures at strain rate of 0.1 s^−1^. It can be seen that as the stress triaxiality increases, the damage value gradually increases. At the boundary of 1000 °C, the critical stress triaxiality, which is a stress triaxiality when the damage value of material reaches the critical damage value under tensile stress state, increases with the increase of temperature. The increase of critical stress triaxiality indicates that the formation and propagation of microcracks in X12 alloy steel need to be carried out under higher stress triaxiality. When the stress triaxiality continues to increase, the crack propagation intensifies, resulting in the material fractures. The increase of temperature leads to the increase of plasticity of X12 alloy steel. Figure 12c shows the equivalent strain and stress triaxiality variation diagram at different strain rates at 1100 °C. It can be seen that the stress triaxiality increases as the equivalent strain rises under the condition of a constant temperature. With the increase of the strain rate, the change rate of stress triaxiality increases gradually. Figure 12d is the stress triaxiality-damage diagram under different strain rates at 1100 °C. It can be seen that with the strain rate increasing, the critical stress triaxiality gradually increases, indicating that the increasing strain rate leads to the increase of the plasticity of X12 alloy steel.

The damage mechanism of X12 alloy steel was further clarified by analyzing SEM photos of tensile specimens. The mechanism of ductile fracture can be attributed to two basic types: tensile ductile fracture (under high stress triaxiality) and shear ductile fracture (under low stress triaxiality). The microscopic morphology corresponding to tensile ductile fracture shows a significant increase in the volume of microvoids and the fracture is covered by holes; the microscopic morphology corresponding to shear ductile fracture shows that a small number of pores can nucleate, and the volume of the pores does not change much, but the hole shape is significantly elongated along the direction of the shear band [30,31,32]. Based on the above analysis, it can be seen that the stress triaxiality in the center of the sample is always positive during stretching process and the material is always in a state of tensile stress, which will promote the generation and propagation of cracks. Figure 13a shows the fracture morphology at 1000 °C, 0.5 s^−1^. It can be seen that the shape of the fracture is like a cup-cone, there are many uneven dimples on the fracture, which are typical ductile fractures. The central area of the fracture is ridge-shaped, with large undulations on the surface. The area around the center is flatter than that in the center, where cracks grow faster. The edge of the sample forms an angle to the center of the sample, where the unstable crack growth occurs [33]. Figure 13b is an enlarged view of the fracture morphology at 1050 °C, 0.1 s^−1^. It can be seen that there are inclusions in the dimples. When plastic deformation occurs, the existence of inclusions will cause stress concentration and high stress triaxiality. The inclusions will hinder the sliding of the matrix, resulting in gradual increase of the stress triaxiality between the matrix and the inclusions. When the stress triaxiality exceeds the critical stress triaxiality, the inclusions will separate from the matrix and form cracks, which gradually expand and lead to the fracture of the material eventually [34,35,36]. Figure 13c,d show the enlarged fracture morphology at 950 °C and 1000 °C at strain rate of 0.5 s^−1^. It can be seen from Figure 12 that when the strain rate is constant, the critical stress triaxiality gradually increases with the increase of temperature, indicating that the formation and propagation of microcracks in X12 alloy steel need to be carried out under higher stress triaxiality. Therefore, with the increase of temperature at the same strain rate, the depth of dimples increases.

## 5. Conclusions

In this paper, the tensile test of X12 alloy steel at temperatures of 950–1150 °C and strain rates of 0.1–5 s^−1^ was accomplished on the Gleeble-1500D. The high temperature damage model of X12 alloy steel was proposed by introducing the influence of temperature and strain rate on damage evolution, on the basis of the Oyane damage model. The parameters of the damage model at elevated temperature of X12 alloy steel were identified by the reverse analysis method. The damage model at elevated temperature was combined to the FORGE^®^ to simulate tensile processes at elevated temperature. The precision of the damage model was verified by contrasting the fracture displacement of the specimen obtained by simulation and experiment, respectively. The results show that it can predict the damage behavior of X12 alloy steel well under hot forging process.

Stress triaxiality is a significant factor affecting the damage behavior of ductile materials. The influence of temperature and strain rate on the stress triaxiality of X12 alloy steel was analyzed. The critical stress triaxiality, which is a stress triaxiality when the damage value of material reaches the critical damage value under tensile stress state, increases with the increase of temperature and strain rate. Through the analysis of SEM photos, it is found that the fracture surface of X12 alloy steel was relatively rough under the action of high stress triaxiality and contains a lot of dimples, in which inclusions can be clearly observed. The results show that the high temperature fracture mechanism of X12 alloy steel under high stress triaxiality is mainly caused by the nucleation, growth, and coalescence of voids.

## Figures and Tables

**Figure 1 materials-14-00695-f001:**
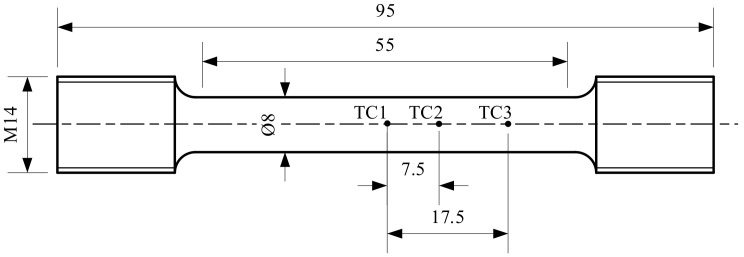
The size of hot tensile specimen and the welding positions of the thermocouple wires [mm].

**Figure 2 materials-14-00695-f002:**
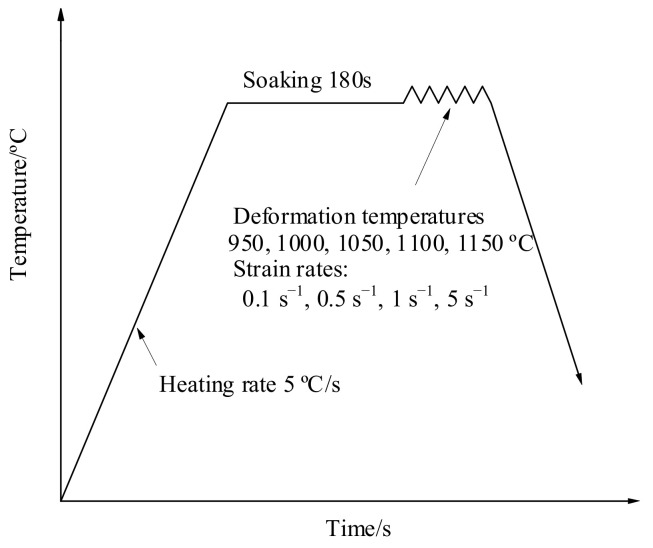
Experimental process of hot tensile tests.

**Figure 3 materials-14-00695-f003:**
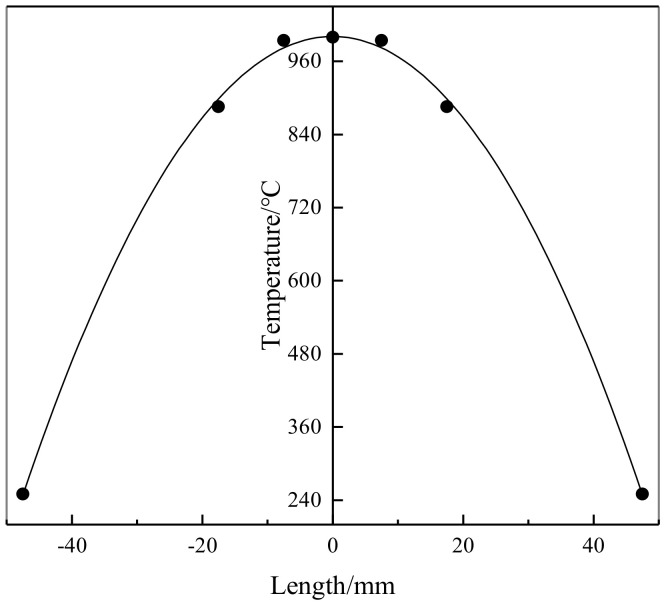
Temperature distribution along length direction of X12 steel at 1000 °C.

**Figure 4 materials-14-00695-f004:**
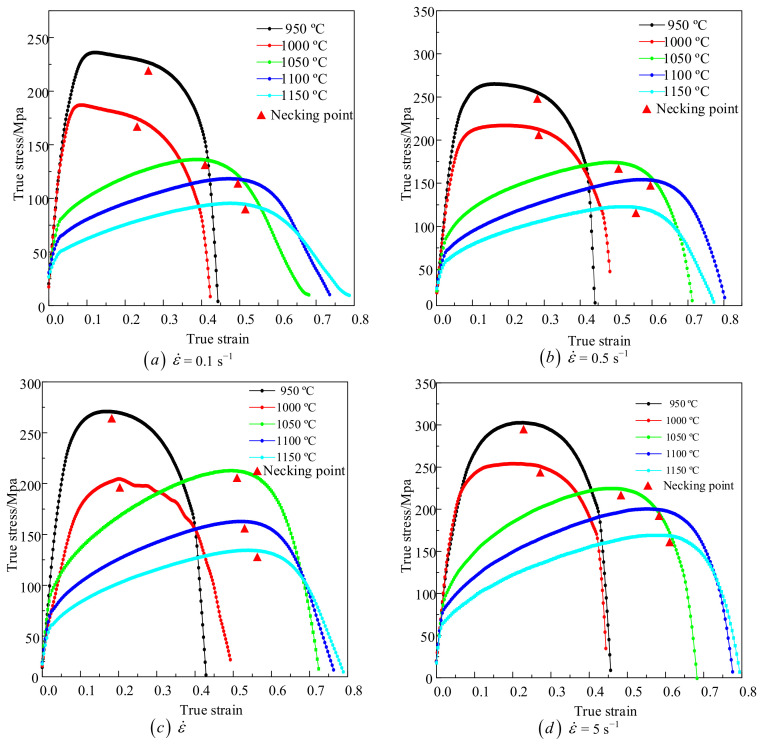
Stress–strain curves from tensile testing at different temperatures and strain rates (**a**) 0.1 s^−1^, (**b**) 0.5 s^−1^, (**c**) 1 s^−1^, (**d**) 5 s^−1^.

**Figure 5 materials-14-00695-f005:**
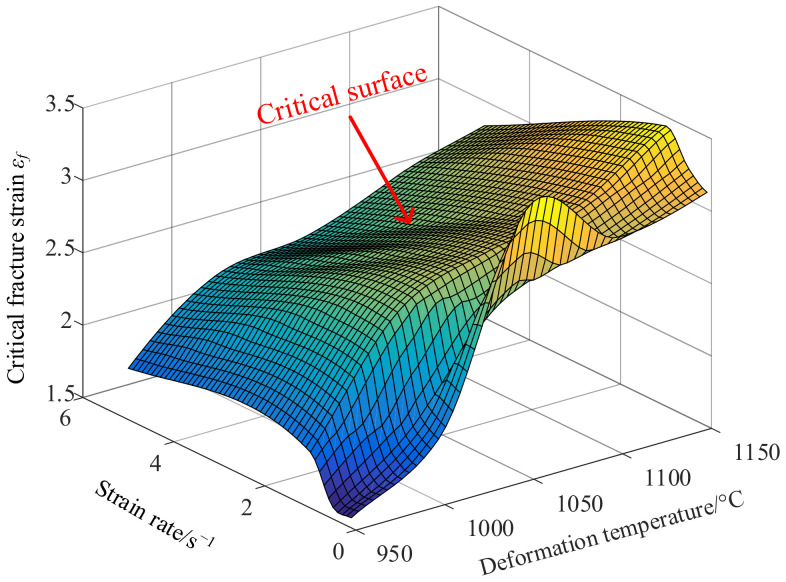
The relationship between critical fracture strain, strain rate, and temperature.

**Figure 6 materials-14-00695-f006:**
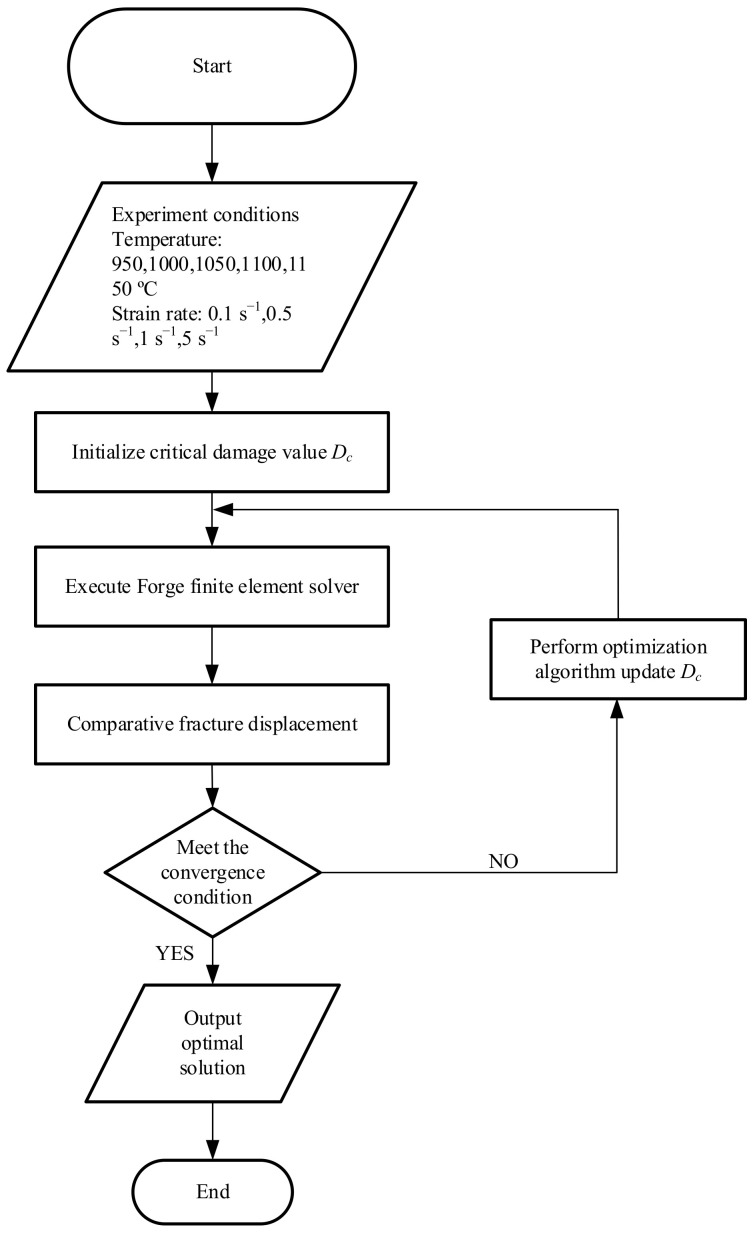
Reverse method flow chart.

**Figure 7 materials-14-00695-f007:**
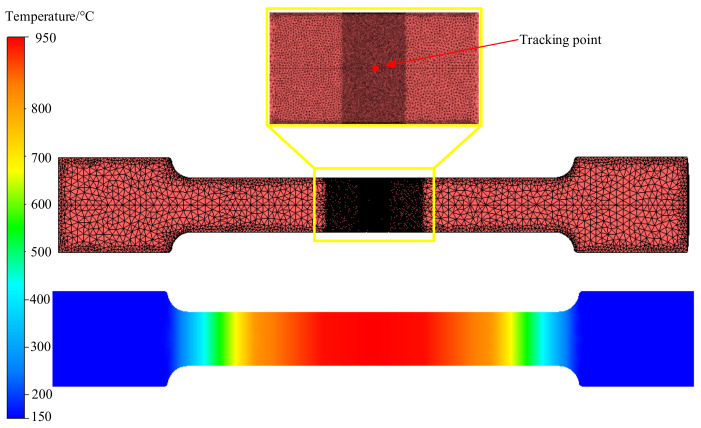
Mesh size distribution and temperature distribution at 950 °C along the sample length direction.

**Figure 8 materials-14-00695-f008:**
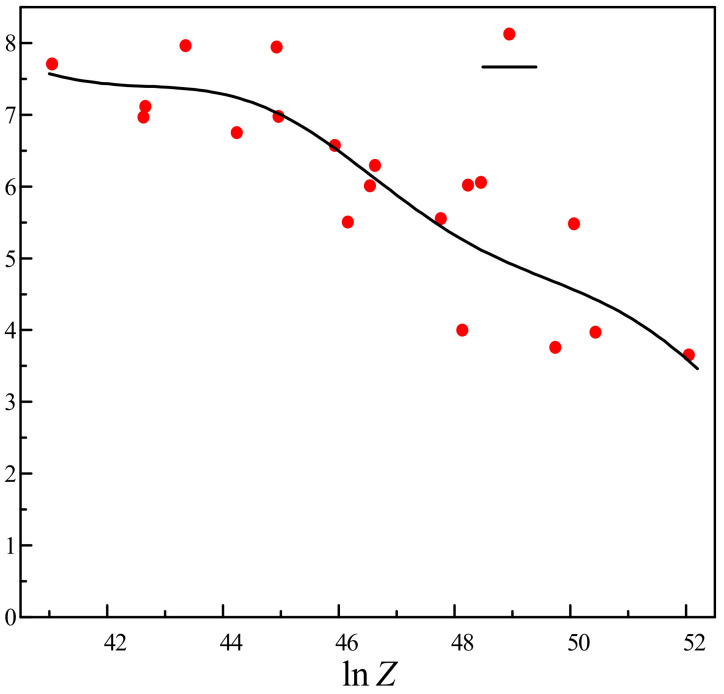
The relationship between lnZ and $D_{c}$.

**Figure 9 materials-14-00695-f009:**
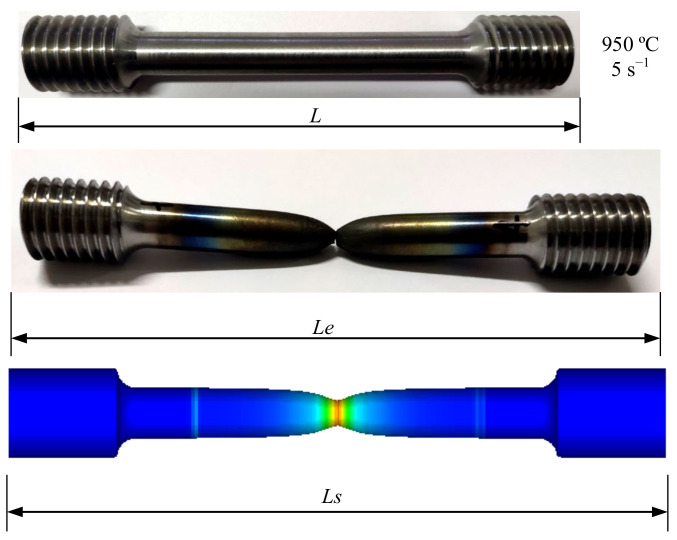
Comparison of simulated fracture displacement and experimental fracture displacement of the sample at 950 °C/5 s^−1^.

**Figure 10 materials-14-00695-f010:**
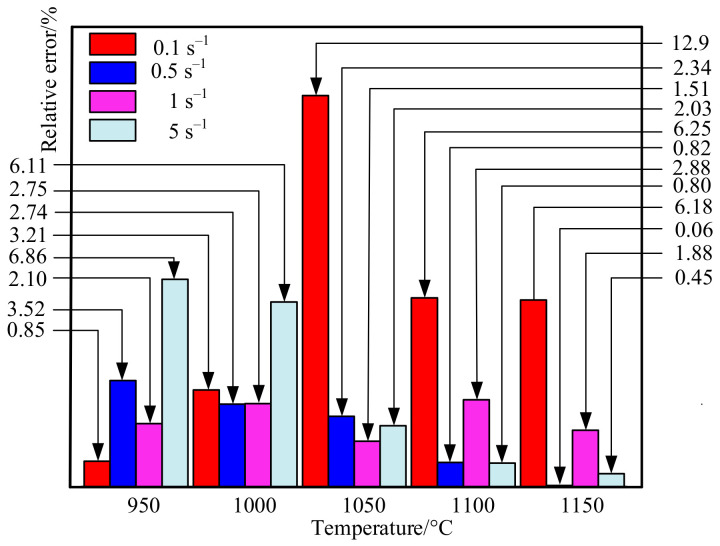
Relative error diagram of X12 steel fracture displacement and sample fracture displacement during simulation.

**Figure 11 materials-14-00695-f011:**
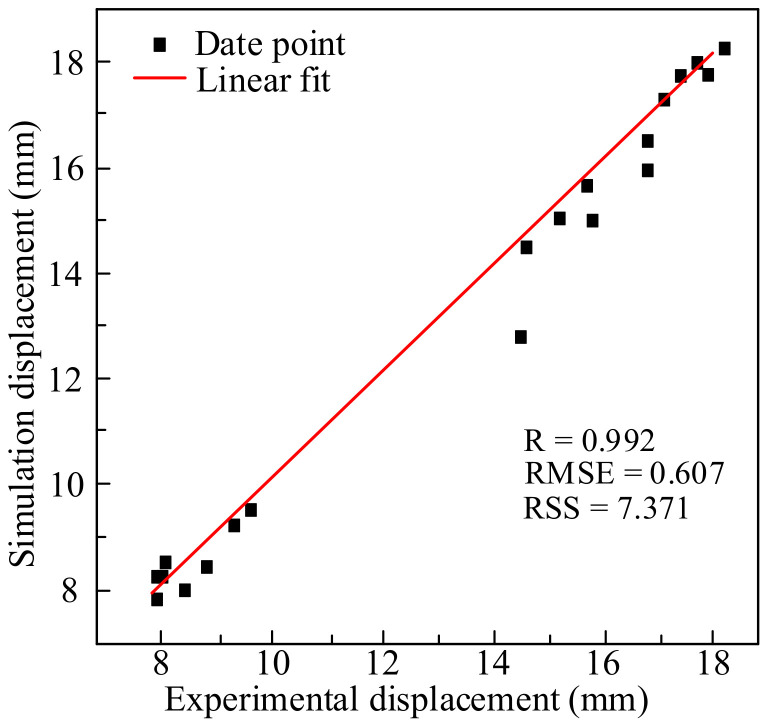
Correlation between simulation and experimental displacement values.

**Figure 12 materials-14-00695-f012:**
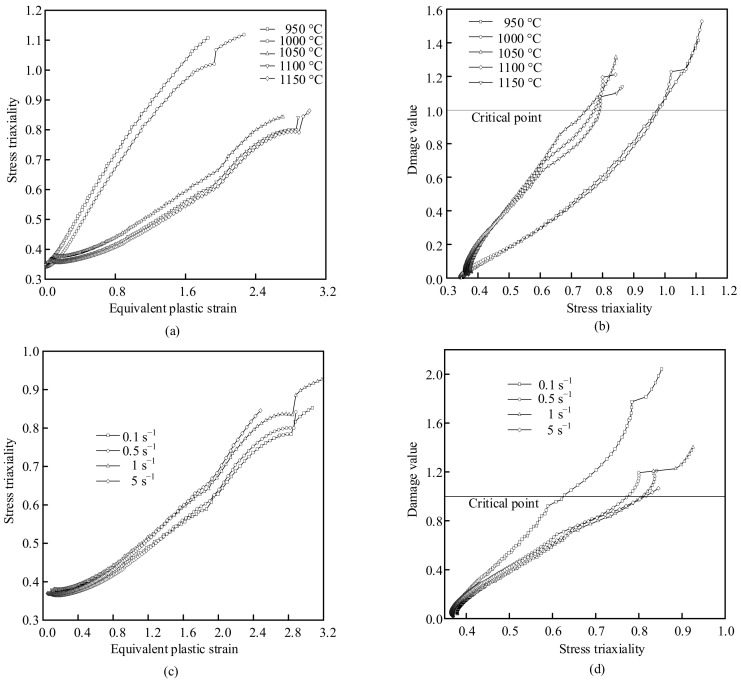
Stress triaxiality and damage changes under different deformation conditions: (**a**) Change of equivalent strain–stress triaxiality at 0.1 s^−1^, (**b**) Stress triaxiality-damage diagram at 0.1 s^−1^, (**c**) Equivalent strain–stress triaxiality at 1100 °C, (**d**) Stress triaxiality-damage diagram at 1100 °C.

**Figure 13 materials-14-00695-f013:**
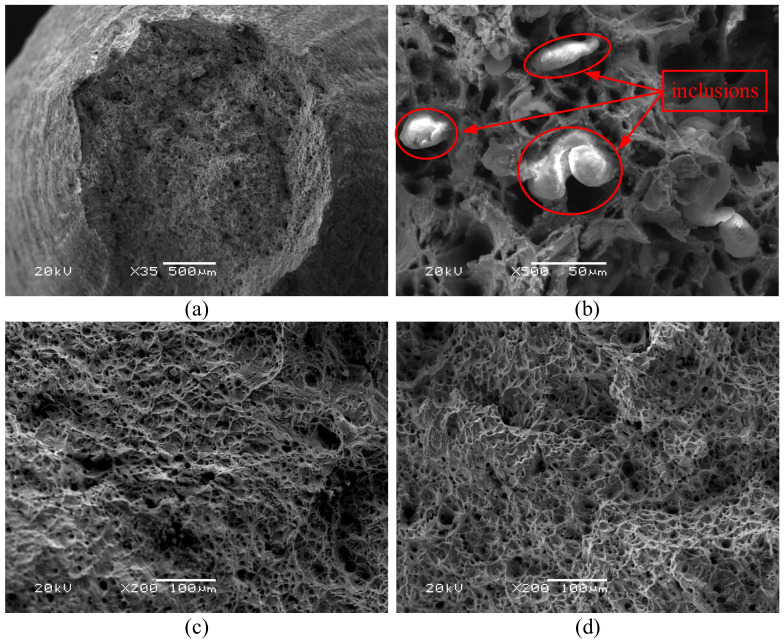
SEM photographs under different deformation conditions (**a**) 1000 °C, 0.5 s^−1^; (**b**) 1050 °C, 0.1 s^−1^; (**c**) 1000 °C, 0.5 s^−1^; (**d**) 1000 °C, 0.5 s^−1^.

**Table 1 materials-14-00695-t001:** Chemical composition of X12 alloy steel (mass percentage: wt.%) [26].

C	Si	Mn	Cr	Mo	W	Ni	V	Nb	N	Cu
0.188	0.104	0.420	11.00	1.029	0.95	0.744	0.207	0.069	0.05	0.186

**Table 2 materials-14-00695-t002:** X12 steel Hansel–Spittel constitutive model parameters.

A	m_1_	m_2_	m_3_	m_4_
23351.9437	−0.00442	0.14754	0.07427	−0.01065

**Table 3 materials-14-00695-t003:** Critical damage values $D_{c}$ of Oyane damage model at different temperatures and strain rates.

Strain Rate/s^−1^	Temperature/K	$D_{c}$	lnZ
0.1 s^−1^	1223	3.9971	48.1379
	1273	5.5027	46.1567
	1323	8.7847	44.3253
	1373	6.9665	42.6273
	1423	7.7040	41.0486
0.5 s^−1^	1223	3.7566	49.7473
	1273	5.5493	47.7662
	1323	6.5694	45.9348
	1373	6.7476	44.2367
	1423	7.1151	42.6580
1 s^−1^	1223	3.9700	50.4405
	1273	6.0561	48.4593
	1323	6.2921	46.6279
	1373	7.9420	44.9299
	1423	7.9597	43.3512
5 s^−1^	1223	3.6500	52.0499
	1273	5.4778	50.0688
	1323	6.0182	48.2373
	1373	6.0061	46.5393
	1423	6.9765	44.9606

## Data Availability

The data presented in this study are available on request from the corresponding author. The data are not publicly available due to these data are part of ongoing research.

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
