# Peer review of "Parameters Identification of High Temperature Damage Model of X12 Alloy Steel for Ultra-Supercritical Rotor Using Inverse Optimization Technique"

_materials, 2021, doi:10.3390/ma14030695_

Round 1

Reviewer 1 Report

The paper is devoted to modeling of fracture of X12 chromium steel under elevated temperatures in the range of strain rates from 0.1 to 5 1/s. The topic is within the scope of the journal and of great interest for the readership because of the specificity of the investigated steel.

However, I suggest the quality of presentation should be improved:

  1. Section 2. should be improved. In particular, the information on the number of specimens tested is missing. Confidence intervals for parameters determined experimentally should be indicated.
  2. Line 205. The statement: “However, X12 alloy steel has the maximum fracture strain when the temperature is 1050 °C and the strain rate is 0.1 s”. Discussion on nonmonotonous behavior of the maximum fracture strain with respect to temperature at the strain rates of 0.1 /s is necessary, because this influences the choice of approximation function (Fig.8.). Information on the dispersion of the experimental values of the maximum fracture strains of steel for specific test conditions should be provided.
  3. Line 210. Fig.4 shows strain-stress curves. The procedure for determining the stress-strain curves should be described.It is not clear from the text of the article that the diagrams of true stresses and true strains or engineering stresses and engineering strains are shown.
  4. Line 242. The Zener-Hollomon model includes the thermodynamic temperature T in an absolute temperature scale.Therefore, T should be measured in Kelvin, but in the manuscript, T is given in Celsius degrees.
  5. Line 358. The statement: “It can be seen from Figure 12 that when the strain rate is constant, the critical stress triaxiality gradually increases with the increase of temperature, which indicates that the formation and propagation of microcracks in X12 alloy steel need to be carried out under higher stress triaxiality.” Does this mean that fracture of X12 alloy steel does not occur at pure shear (the stress triaxiality factor=0)? The explanation of term the critical stress triaxiality is necessary.
  6. Comparison of experimental results with the literature is needed.

Reviewer 2 Report

The proposed paper deals with engineering interesting problem.

However, many things need to be improved.

  1. Abstract: target, methods and results
  2. Introduction: to long
  3. English  Language need to be seriously improved
  4. Chapter 2: Not enough explanations
  5. Stress-strain curves: none curve is ok, since stress-strain curve need to be carried out in such a way that minimum three strain rate are used: elastic region, yielding and third part (necking) , every part need to have different strain rate if you want to obtain real  curve. Use any standard to take appropriate  strain rate for  mentioned period of the process.
As far as can be seen, it is not about the materials but about the material. Furthermore, what are the methods. Only one procedure was mentioned. In addition, it is to be expected that this procedure will be represented in its entirety by a curve that is not the same as any of the curves in Figure 4, because these are not real stress-strain curves. Namely, the real stress-strain process has several strain rates to be as close as possible to the real process, that is, the deformation rate passes through the elastic region, the second strain rate is in the flow region or elastoplasticity, and finally, this one, in the last part of the diagram . By the way, all stress-strain curves shown with constant strain rates in Figure 4 are only figurative but in reality inaccurate and unnecessary. They can only be used to compare the magnitudes of mechanical properties that depend on the degree of deformation.

Round 2

Reviewer 1 Report

As the objections have been adressed, i recommend the article for presentation.

Author Response

Dear reviewer

Thank you very much for reviewing my manuscript

Reviewer 2 Report

Some things in the manuscript are improved.

Other comments are as follows:

  1. The process of high temperatures is considered, and where is the impact of creep?
  2. Existing software used, what's new?
  3. Despite the clarification on the use of a constant value of the strain rate and the claim that the differences are small, the answer is neither correct nor sufficiently accurate.
  4. Each stress-strain diagram has certainly 3 strain rates that are closest to the real process. Using references from the literature to cover what has been done is not an accurate thing; that is, it is an approximation. Mention which three strain rates were used in the process run. We can say it won’t work or anything like that, but we can’t claim it’s true. Subject to the basics of Figure 4, shows that the sizes of the properties  change with different deformation rates. Therefore, either say that one speed was used, or state which three strain rates  are used in one process, or considered process is not correct.
  5. Example of inaccuracy in language: check the text. Some sentences are too long and poorly conceived.

Round 3

Reviewer 2 Report

The authors tried to answer the questions only formally. Apart from a small, yet insufficient language correction, the rest has not been made.
In addition to the effort made to provide some explanations, no valid additional experiment has been made as valid answere to the questions.
First of all, the influence of high temperature on the stress-strain response and then the creep response was considered. As for the response of the material (stress-strain) at elevated temperatures clearly is that the diagrams shown with constant strain rate in the experiment can serve only for the purpose of mutual comparison, but not the accuracy of an individual experiment. Such clarifications should be stopped. The experiment requires three strain rates during the process.
Another explanation related to creep can be found even on the Internet, and the one in the shorter or longer textual explanation does not have any special significance. It should be noted here that it is possible to determine the causes of creep deformation by analyzing the missing microstructure. The microstructure that is shown does not refer to creep. Third, when considering the occurrence of creep in the process of machine / element operating at a constant load is not the same as the stress - strain response of the process to fracture viewed separately. During the loading of the element / machine as a dynamic process it can be considered in combination with creep, but this is not done here because it is not taken even as the subject of consideration. The title of the paper refers to the consideration of the material in use for a rotor that will work in high temperature conditions, ie in a combination of fatigue and creep, ie creep - fatigue case. There is nothing about this here. Anyway, I think that a lot of effort has been put into the work, but that it has not been completed, in places made with omissions and incomplete, even from the position of the title itself. It needs improvement and even a title change in the sense of knowing what will be considered. Finally, it is not the intention to complicate things but to improve them. If the editor decides otherwise, it will be so.

Author Response

Dear reviewer:

Thank you very much for your review and we sincerely appreciate you for your valuable comments on our article, especially for pointing out our problems in the stress-strain curves. Your opinion makes us realize that there is still no guarantee that the test will be performed at the set strain rate throughout the process of test, though the the Gleeble testing machine is used. The strain rates in elastic, plastic and necking stages are different, this time we revised the manuscript and added the description in this aspect (as you may see in the yellow text highlight color in Page 4(171-184 lines)). As for the creep problem you put forward, although adding the research on creep will make our research more perfect, this does not meet our original purpose of the paper. The purpose of our research is to establish a damage model considering temperature and strain rate and predict the evolution of damage during rotors manufacturing. To illustrate the purpose of our paper, we have revised the manuscript(as you may see in the green text highlight color). Finally, thank you again for reviewing our manuscript.